# Meta-Analyses of the Relationships between Family Systems Practices, Parents’ Psychological Health, and Parenting Quality

**DOI:** 10.3390/ijerph20186723

**Published:** 2023-09-07

**Authors:** Carl J. Dunst

**Affiliations:** Orelena Hawks Puckett Institute, Asheville, NC 28730, USA; cdunst@puckett.org

**Keywords:** family needs, family resources, family supports, family strengths, parents’ psychological health, parenting beliefs, parent involvement, parenting practices, meta-analysis

## Abstract

(1) Background: Family systems theories include assertations that both personal and environmental factors are determinants of parents’ psychological health, well-being, and parenting quality. Applied family systems theories focus on determinants that can be operationalized as intervention practices. The analyses described in this paper focused on the direct and indirect effects of four family systems practices (family needs, resources, supports, and strengths), parents’ psychological health (depression, well-being, etc.), and parenting quality (parenting beliefs, involvement, and practices) in families of children with identified disabilities, medical conditions, or at-risk conditions for poor outcomes; (2) Methods: Data from previously completed meta-analyses of the relationships between family systems practices and parents’ psychological health outcomes and parenting quality outcomes were reanalyzed. Next, a meta-analysis of the relationships between parents’ psychological health and parenting quality was completed to identify which predictors were related to which parenting quality outcomes. Both main effects and mediated effects were examined; (3) Results: The four family systems practices were each related to six different psychological health measures and three parenting quality measures. The six different parental psychological health measures were also related to the three parenting quality measures. The relationships between family systems practices and parenting quality were partially mediated by parents’ psychological health; (4) Conclusions: The effects of family systems practices and parents’ psychological health on parenting quality were primarily direct and independent. The relationships between family systems practices and parenting quality were partially mediated by parents’ psychological health. Future research should focus on the identification of other mediator variables found to be important for explaining the indirect effects of family systems practices measures on parenting beliefs, behavior, and practices.

## 1. Introduction

Parenting quality is a multidimensional construct that includes parenting beliefs, parental involvement in children’s learning, and parenting practices to promote child learning and development [1,2,3,4,5]. Parenting beliefs include appraisals of parenting competence [6] and beliefs that parenting actions will have desired consequences [7]. Parental involvement includes parents’ effort to engage their children in everyday learning activities and opportunities [8,9] and participation in children’s formal schooling [10,11]. Parenting practices include caregiver emotional warmth, responsiveness, guidance, and other supportive actions to encourage and reinforce child learning and development [12,13].

Decades of research find that parenting quality matters a great deal in terms of positive child well-being, learning, and development [14,15,16,17]. Parenting, however, includes both ups and downs and pleasures and pains [18,19]. Nelson et al. [18] describe how poor psychological health has negative consequences on parenting quality and how positive psychological health has positive consequences on parenting quality. Research reviews and meta-analyses of parenting beliefs, behavior, and practices include evidence that parents’ psychological health is an important factor in explaining differences in parenting quality [20,21,22,23]. Rueger et al. [23], for example, found in a meta-analysis of parenting studies that positive well-being was associated with positive parenting practices and that negative well-being was associated with poor parenting practices.

The birth and rearing of a child with special needs (developmental disability, chronic health condition, etc.) or raising a child under adverse conditions (poverty, single parenting, etc.) are often associated with increased parenting stress and burden (e.g., [24,25,26,27,28]). Parents’ poor psychological health, in turn, has been found to have adverse effects on parenting quality (e.g., [21,29,30]). However, like parents of children without special needs or adverse life circumstances, parents of children with disabilities, chronic conditions, or adverse life conditions vary considerably in terms of parenting quality [31,32].

### 1.1. Systems Theories and Parenting Quality

Both personal and environmental factors have been identified as determinants of variations in the psychological health, well-being, and parenting quality of parents of children with and without disabilities (e.g., [33,34,35,36]). A number of systems theories include hypotheses about the factors associated with the relationships between parents’ psychological health and parenting quality and the factors associated with variations in both of these parenting characteristics [37,38,39,40]. Bronfenbrenner [41], for example, noted that parents’ abilities to effectively carry out child-rearing responsibilities depend upon the role demands, stressors, supports, and resources available in different ecological settings.

Systems theories have been used by both researchers and practitioners to understand parent and family member reactions and adjustments to the birth and rearing of a child with a disability or medical condition or a child who is at risk for poor outcomes [42,43,44]. Algood et al. [42], for example, used Bronfenbrenner’s ecological systems theory to review the literature in terms of the personal and environmental factors that are associated with parenting successes and challenges. Allgood [42] also described how these factors are related to parents’ psychological health and parenting practices.

#### Applied Family Systems Theory

The author and his colleagues have used different systems theories to develop an applied family systems model for both research and practice in families of children with developmental disabilities or delays, families of children with different special health care needs, and families with children who are at risk for poor outcomes for family-related risk factors. The model has been revised and updated based on results from research studies and lessons learned from everyday practice with parents and their children (see [45,46]). The model differs from other family and social systems models by focusing on theoretical constructs that can be operationalized as intervention practices [47]. Nonintervention variables such as personal characteristics are examined as potential moderator variables for explaining differences in the relationships between intervention practices and outcomes of interest. These include variables such as child disability status, parent age and education, and family socioeconomic status.

Figure 1 shows the four components of the applied family systems model: family- identified needs; the social supports and resources for meeting needs; the use of family and family member strengths to procure needed supports and resources; and practitioner use of family-centered help-giving practices to strengthen and build parent and family member capabilities to engage in desired child, parent, and family activities [46]. The intervention model is implemented by a practitioner using family-centered capacity-building practices to facilitate family member identification of (1) unmet needs, (2) the resources and supports for needs satisfaction, and (3) the use of family and family member strengths to obtain needed resources and supports. Markers for the effectiveness of the family systems intervention practices include parent and family member self-efficacy beliefs about the ability to execute courses of action to both meet needs and achieve goals and aspirations [48]. The desired outcomes of the family systems intervention practices are improved parent, family, and child psychological health and well-being, positive family member interactions, parent use of development-enhancing parenting practices, and child learning and development.

Considerable effort has been expended by the author and his colleagues in investigating the relationships among the family systems intervention practices and parent, family, and child outcomes. This has included meta-analyses of practitioner use of family-centered practices [49,50], family needs [51], the sources and types of family social supports [52,53,54], the adequacy of family resources [55,56,57], and family strengths [58,59,60]. The meta-analyses all involved tests of the hypothesized relationships between the Figure 1 model components and different parent, family, and child outcomes.

Results from the meta-analyses showed that large numbers of unmet family needs are associated with poorer parent psychological health and well-being, but that family-centered practices, family resources, social supports, and family strengths are associated with attenuated poor parent psychological health and enhanced parent well-being. Moderator analyses found that the sizes of effects for the relationships between the family systems practices and parents’ psychological health were much the same for parents of children with identified disabilities, developmental delays, special health care needs, and children at risk for poor outcomes [51,53,55,57,60], respectively, with only a few exceptions [52,56].

A number of meta-analyses include findings regarding the relationships between the family systems intervention practices and parenting quality [51,52,56,57,58,60]. A large number of unmet family needs are related to more negative parenting beliefs. In contrast, family-centered practices, family resources, social supports, and family strengths were associated with both increased parental involvement in their children’s learning and education and more positive parenting practices.

The findings in the different research syntheses of the family systems intervention practices studies are all consistent with the basic premises of the applied family systems intervention model [46]. There are, however, several caveats to the methods and results that need to be mentioned to place the findings in both conceptual and methodological contexts. First, the categorization of the psychological health measures in the different meta-analyses varied for different reasons (e.g., the purpose of the meta-analyses; the health and well-being measures used in the primary studies). Second, the same was the case for the parenting quality measures, where the categorization of parenting measures often varied. For example, in some studies the investigators did not differentiate between parenting beliefs and parenting practices but rather treated them as the same constructs. For both reasons, the results in the different meta-analyses cannot be compared to ascertain which family systems practices are related to which psychological health or parenting quality measures.

These methodological differences were addressed in this study by reorganizing, recoding, and reanalyzing the data in the meta-analyses so that the categorization of the psychological health measures (e.g., depression, well-being, caregiving burden) and parenting quality measures (parenting beliefs, parental involvement, parenting practices) were the same in each meta-analysis, which was the focus of additional analyses described in this paper. This also involved a meta-analysis of the relationships between parents’ psychological health and parenting quality so that the data necessary to conduct meta-analytic mediational analyses could be performed [61].

### 1.2. Aims of the Study

This study addresses both the “lack of understanding regarding the [relationships] between specific dimensions of parental mental health and parenting” and the mechanism for understanding if and how intervention-related variables are directly or indirectly related to parenting quality, as described in the call for papers for the Special Issue of IJERPH on Parenting and Mental Health. Accordingly, correlational meta-analyses were used to test the hypothesized relationships between the family systems practices measures and the psychological health and parenting quality measures. This type of meta-analysis uses the correlations between variables of interest to determine the strength of relationships between the independent and dependent variables; specifically, weighted average correlation coefficients provide the best estimates of population effect sizes between measures.

The intervention-related variables that were the focus of investigation included family needs, family resources, family supports, and family strengths, as well as their relationships with parents’ psychological health and parenting quality. The relationships between six different psychological health measures (general psychological health, depression, psychological stress, well-being, caregiving burden, and parenting stress) and three different parenting quality measures (parenting beliefs, parental involvement, and parenting practices) were also a focus of investigation.

The specific aims of the study were to:Ascertain the relationships between family systems intervention practices and parents’ psychological health [41].Ascertain the relationships between family systems intervention practices and parenting quality [40].Ascertain the relationships between parents’ psychological health and parenting quality [20].Ascertain the indirect relationships between family systems intervention practices and parenting quality mediated by parents’ psychological health [62].Ascertain whether the relationships between family systems intervention practices and parenting quality are partially, completely, or not at all mediated by parents’ psychological health [63].Ascertain if the targets of appraisals of the psychological health measures (parent-focused vs. nonparent-focused) differentially influenced the mediated effects between family systems practices and parenting quality [64].

The results were expected to identify which family systems intervention practices measures and which psychological health measures proved most important in terms of explaining variations in parenting beliefs, parental involvement, and parenting practices. The results were also expected to ascertain if the hypothesized relationships between the family systems intervention practices and parents’ psychological health and parenting quality were supported by the findings in the studies included in the meta-analyses.

## 2. Methods and Materials

### 2.1. Approach

The guidelines described by Siddaway et al. [65] for conducting a systematic review and meta-analysis were used in each of the family systems practices meta-analyses, as well as the meta-analysis described in this paper for the relationships between parents’ psychological health and parenting quality. Meta-Essentials was used to compute the average weighted effect sizes between the independent and dependent measures using random effects models [66,67]. The methods described by Kenney [63] were used to perform the mediated analyses. The Sobel test was used to determine if the relationships between family systems practices and parenting quality were mediated by parents’ psychological health [68]. The American Psychological Association reporting standards were used to describe the results of each meta-analysis [69].

### 2.2. Search Strategy

The search sources for the studies in each of the family systems practices meta-analyses were PsycNet, ProQuest Central, PubMed, ERIC (Educational Resource Information Center), Google Scholar, Directory of Open Access Journals, Bielefeld Academic Search Engine, and CORE. Controlled vocabulary searches were used in PsycNet, ProQuest Central, PubMed, and ERIC. Natural language searches were used in all eight search engines. The same sources were used to locate studies of the relationships between parents’ psychological health and parenting quality.

The search terms for locating studies in each of the family systems practices meta-analyses are described in detail in the research reports. This included the names of the scales used to measure each of the systems practices found in the literature and the terms used to describe each family systems practice construct also found in the literature. The search terms for locating the psychological health measures studies included the names of the scales in the family systems practices meta-analyses, the names of other psychological health measures, and the terms used to describe different types of psychological health constructs found in the literature. These included, but were not limited to, general psychological health, depression, psychological stress, psychological distress, stressors, stressful life events, anxiety, well-being, quality of life, parenting stress, parenting distress, caregiving burden, and caregiver burden. The search terms for the parenting quality studies included the names of the scales in family systems practices meta-analyses and the terms used in the literature for describing different types of parenting beliefs, parenting behavior, and parenting practices. These included, but were not limited to, parent self-efficacy, parenting self-efficacy, parenting beliefs, parenting competence, parental involvement, parental engagement, parent-child activities, parenting practices, parent-child interactions, and parent-child relationships.

### 2.3. Inclusion and Exclusion Criteria

Studies were included in the family systems practices meta-analyses if self-report measures of the family systems practices of interest were completed by the study participants and the correlations with either or both psychological health and parenting quality measures were reported. Studies were also included if self-report measures of one or more psychological health measures were completed by the study participants and the correlations with any of the three parenting quality constructs were reported. The study participants were parents or other primary caregivers (e.g., grandmothers raising grandchildren) of children with identified disabilities, special health care needs, or children at risk for poor outcomes from birth to 18 years of age. No limitations were placed on the type of research report, where the studies were conducted, or the year of publication.

Studies were excluded if only significant correlations between measures or incomplete correlations were reported, the study participants were not parents or primary caregivers of one of the three targeted groups of children and adolescents, the parents or primary caregivers had a diagnosis of a mental health or medical health condition, or insufficient information was included to ascertain the direction of effects between the independent and dependent measures. Studies were also excluded in the meta-analysis of the relationships between the psychological health and parenting quality measures if they were conducted during the COVID-19 pandemic, since the studies in the family systems practices meta-analyses were conducted before that adverse event. This exclusion was chosen so as not to add confounds to the results reported in this paper.

### 2.4. Study Measures

The meta-analyses used in this study included family systems practices measures, psychological health measures, and parenting quality measures. The complete list of scales in each family systems meta-analysis can be found in the research reports described above. The psychological health and parenting quality measures in the meta-analysis completed for this study can be found in the Appendix A for this study.

#### 2.4.1. Family Systems Measures

The family systems practices were measured using different family needs scales [70,71], social support scales [72,73], family resources scales [74,75], and family strengths scales [76,77]. The measures cited for each family systems practice are the ones most often used in the studies in the meta-analyses.

#### 2.4.2. Psychological Health Measures

Six different psychological health constructs were the focus of investigation in the different family systems practices meta-analyses, as well as the meta-analysis completed for this paper. The constructs and associated measures included general psychological health [78,79,80], depression [81,82], stress [83,84], well-being [85,86], caregiving burden [87,88], and parenting stress [89,90,91]. The general psychological health, depression, stress, and well-being scales all assessed psychological health without reference to a child with an identified disability, medical condition, or at-risk status. In contrast, the caregiving burden and parenting stress scales all measured psychological health in reference to the children’s identified conditions or at-risk status [64].

#### 2.4.3. Parenting Quality Measures

Three types of parenting quality measures were the focus of investigation: parenting beliefs, parental involvement, and positive parenting practices. Parenting beliefs included parents’ attitudes toward childrearing [92], parents’ sense of competence [93], and parenting self-efficacy appraisals [94]. Parental involvement included parents’ efforts to engage their children in everyday child learning activities [95,96] and parents’ involvement in their children’s formal early childhood intervention and education [97,98,99]. Positive parenting practices included behavior used to promote informal and formal child learning, development, and education [9,99,100,101].

### 2.5. Data Preparation

The zero-order correlations between the different sets of measures and associated sample sizes were inputted into Meta-Essentials [66] in order to perform the meta-analyses. In some cases, the family systems meta-analyses data were reanalyzed so that the psychological health measures and parenting quality measures were all categorized in the same manner, as described above. The measures for the relationships between the psychological health and parenting quality measures were categorized in the same manner as the family systems practices meta-analyses.

Preliminary analyses were conducted for each set of measures to identify outliers and influential cases [102]. An iterative process was used to delete cases. Effect sizes that were both outliers and influential cases were deleted first, and then the analyses were rerun to determine if additional cases needed to be deleted. If other effect sizes were still outliers, these were deleted as well, and the analyses were once again rerun. This process was repeated until all outliers and influential cases were removed from the datasets. Any one study could include an effect size that was deleted for one pairwise set of measures but included an effect size for another pairwise set of measures that was not an outlier.

### 2.6. Data Analysis

Both main and indirect effect analyses were conducted to achieve the aims of the study. Three sets of analyses were conducted to test the main effects of the relationships between the study variables. The first set of analyses evaluated the relationships between the four family systems practices measures (needs, resources, supports, and strengths) and the six psychological health measures (general health, depression, stress, well-being, parenting stress, and caregiving burden).

The second set of analyses evaluated the relationships between the four family systems practices measures and the three parenting quality measures (beliefs, involvement, and practices). The third set of analyses evaluated the relationships between the six psychological health measures and the three parenting quality measures.

The fourth set of analyses evaluated the extent to which the relationships between the family systems practices measures and the parenting quality measures were mediated by the psychological health measures. This was determined by the product of the average sizes of effect between the family systems intervention practices and the psychological health measures and the average sizes of effect between the psychological health and parenting quality measures.

## 3. Results

Table 1 shows selected characteristics of the meta-analyses that investigated the relationships between family systems practices, psychological health, and parenting quality measures. The studies in the family systems meta-analyses are included in each of the research reports cited above and referenced in Table 1.

Most of the family systems meta-analyses included between 30 and 82 studies and 3303 to 7675 participants. The parents’ psychological health meta-analysis included 108 studies and 21,784 participants. The studies in all of the meta-analyses were found in both peer-reviewed journal articles and non-peer-reviewed sources. The latter included primarily doctoral dissertations and master’s theses. The primary caregivers in most studies were mothers of children with identified disabilities/developmental delays, medical conditions, or children at risk for poor outcomes. The children ranged in age from less than one year to 18 years of age. The average child age ranged between 6 and 12 years.

**Table 1 ijerph-20-06723-t001:** Selected characteristics of the meta-analyses of the predictors of parents’ psychological health and parenting quality.

	Number of:			Percentage:				
Meta-Analyses	Studies	Participants	Countries	PR Articles	Mothers	Cw DD	Cw MC	Cw AR
Family Needs								
Dunst [51]	31	4543	15	71	81	81	19	0
Family Supports								
Dunst [52]	82	7675	12	57	79	55	19	11
Dunst [53]	51	4540	6	61	87	48	14	27
Dunst [54]	29	3440	10	48	81	90	10	0
Family Resources								
Dunst [55]	50	8183	6	52	81	34	26	40
Dunst [56]	30	5247	4	60	83	46	23	18
Dunst [57]	14	3030	2	57	76	64	36	0
Family Strengths								
Dunst et al. [58]	33	7065	12	42	75	41	0	37
Dunst [59]	14	3491	10	36	85	33	33	15
Dunst [60]	53	4418	9	62	72	38	35	25
Psychological Health								
Dunst [IJERPH]	108	21,784	19	67	88	55	9	28

Notes, Cw = Children with, DD = Developmental disabilities/developmental delays, MC = Medical conditions, and AR = At risk for poor outcomes. PR = Peer-reviewed journal articles. IJERPH = Meta-analysis of the relationships between parents’ psychological health and parenting quality prepared for the special issue of IJERPH on Parenting and Mental Health. Mothers include biological mothers, stepmothers, adoptive mothers, and foster mothers.

### 3.1. Family Systems Practices Effects

The relationships between the four family systems practices and the six different psychological health measures are shown in Table 2. All of the average weighted sizes of effect differed significantly from zero, as evidenced by confidence intervals not including zero [66].

The direction of effects was as expected. A greater number of family needs were associated with poorer general psychological health, depression, stress, caregiving burden, and parenting stress, as well as less positive psychological well-being. In contrast, the presence of more family resources, more social supports, and more family strengths was associated with attenuated poor psychological health and enhanced psychological well-being.

Inspection of the sizes of effect between the family systems practices and the psychological health measures show that effect sizes are much the same for family needs, family resources, and family strengths. In contrast, the sizes of effects between the family supports and psychological health measures are much smaller but still significantly different from zero.

Table 3 shows the sizes of effects between the four family systems practices measures and the three parenting quality measures. All four sets of average weighted sizes of effect differed significantly from zero, as evidenced by confidence intervals not including zero [66]. The direction of effects was as expected. A greater number of family needs were associated with more negative parenting beliefs, less parental involvement in children’s informal and formal learning activities, and less frequent use of positive parenting practices. In contrast, the presence of more family resources, more social supports, and more family strengths was associated with more positive parenting beliefs, more parental involvement in their children’s informal and formal learning activities, and more frequent use of positive parenting practices.

Inspection of the sizes of effect for the relationships between the family systems practices and parenting quality measures shows that the effect sizes for family strengths are larger than those for the other family systems practices measures and are almost twice as large as those for family supports. These results, together with those in Table 2, point to the relative importance of family strengths as a covariate of parents’ psychological health and parenting quality.

### 3.2. Psychological Health Effects

The relationships between the six psychological health measures and the three parenting quality measures are shown in Table 4. The average weighted sizes of effect all differ significantly from zero, as evidenced by confidence intervals not including zero. Poorer general psychological health, depression, stress, caregiving burden, and parenting stress were related to more negative parenting beliefs, less parental involvement in children’s informal and formal learning activities, and less frequent use of positive parenting practices. In contrast, positive psychological well-being was associated with more positive parenting beliefs, more parental involvement in children’s informal and formal learning activities, and more frequent use of positive parenting practices.

**Table 4 ijerph-20-06723-t004:** Average weighted effect sizes (r) for the relationships between the psychological health measures and parenting quality.

Psychological Health Measures	k	N	r	95%/CI
General Health				
Parenting Beliefs	7	943	−0.28	−0.24, −0.33
Parent Involvement	8	1614	−0.18	−0.13, −0.22
Parenting Practices	9	1174	−0.31	−0.20, −0.41
Depression				
Parenting Beliefs	10	2171	−0.41	−0.34, −0.47
Parent Involvement	13	4759	−0.18	−0.15, −0.21
Parenting Practices	20	3079	−0.23	−0.19, −0.26
Stress				
Parenting Beliefs	11	1348	−0.40	−0.35, −0.44
Parent Involvement	7	2271	−0.21	−0.12, −0.29
Parenting Practices	6	1037	−0.30	−0.14, −0.44
Well-Being				
Parenting Beliefs	8	919	0.42	0.33, 0.50
Parent Involvement	5	2533	0.27	0.16, 0.37
Parenting Practices	9	1548	0.20	0.11, 0.29
Caregiving Burden				
Parenting Beliefs	12	1404	−0.34	−0.29, −0.38
Parent Involvement	6	398	−0.24	−0.17, −0.30
Parenting Practices	11	1922	−0.31	−0.23, −0.38
Parenting Stress				
Parenting Beliefs	16	3454	−0.44	−0.39, −0.48
Parent Involvement	10	3388	−0.21	−0.18, −0.24
Parenting Practices	13	3414	−0.30	−0.24, −0.36

Inspection of the sizes of effects between the psychological health and parenting quality measures shows that the effect sizes between depression, stress, well-being, parenting stress, and parenting quality are larger for parenting beliefs compared to parental involvement and parenting practices. In contrast, the sizes of effect between general psychological health, caregiving burden, and the three parenting quality measures were much the same.

### 3.3. Mediated Effects

The sizes of effects between the family systems practices, psychological health, and parenting quality measures were considered the best estimates for determining if the psychological health measures mediated the relationships between the family systems practices and parenting quality measures. For each of the family systems practices measures, two sets of psychological health measures were used in the mediated analyses: the aggregated effect sizes for the four nonparent-focused measures (general health, depression, stress, and well-being) and the two parent-focused measures (parenting stress and caregiving burden). This permitted a determination of whether the targets of appraisals of the psychological health measures [64] influenced the indirect relationships between the family systems practices and parenting quality measures.

Table 5 and Table 6 show the effects decomposition for the direct, indirect (mediated), and total effects between the family systems practices and parenting quality measures mediated by the nonparent-focused and parent-focused psychological health measures, respectively. All of the mediated effects differed significantly from zero, but most accounted for only a small amount of variance in the relationships between the family systems practices and parenting quality measures. (In most cases, the standard errors for the effect sizes were very small, which resulted in the Sobel Tests yielding statistically significant effect sizes.)

A comparison of the mediated effects in Table 5 and Table 6 shows that sizes of effect for the non-parent-focused and parent-focused psychological health measures are much the same for the relationships between family needs, family resources, family strengths, and parenting beliefs measures. The same is the case for the relationships between family needs, resources, strengths, and parenting practices measures. In contrast, the mediated effects for the relationships between the family supports measures and both parenting beliefs and parenting practices measures were notably smaller.

Examination of the total effects for the family systems measures shows that the sizes of effect for the family strengths measures are 40 or larger for all three parenting quality measures and are nearly the same for both the parent-focused and nonparent-focused psychological health measures. The only other total effect sizes that are 40 or larger are for the relationships between family needs and parenting beliefs for both the parent-focused and nonparent-focused psychological health measures, as well as for the relationship between family resources and parenting practices for the parent-focused psychological health measures.

## 4. Discussion

Results from the secondary meta-analyses showed that the four family systems practices were all related to the six parents’ psychological health measures (Aim 1). The results also showed that both the family systems practices and the parents’ psychological health measures were related to the three parenting quality measures (Aims 2 and 3). These results are consistent with the foundations of the applied family systems theory that guided the conduct of meta-analyses [46]. This pattern of results is also consistent with Bronfenbrenner’s [41] assertations that parents are not able to carry out child-rearing responsibilities without adequate supports and resources that provide them the time and psychological energy to engage their children in development-enhancing learning activities and employ positive parenting practices.

The results from the mediated analyses showed that the relationships between the family systems practices and parenting quality measures were partially mediated by parents’ psychological health (Aims 4 and 5) but accounted for only small amounts of variance between measures. The comparisons between the nonparent-specific and parent-specific psychological health measures showed that caregiving burden and parenting stress mediated somewhat more of the variance for the relationships between the family systems practices and parenting quality measures than did the general psychological health measures (Aim 6). This pattern of results is similar to that found in other studies where parent-focused but not nonparent-focused parenting stress mediated the relationships between family supports and resources and parenting quality [103,104]. Bonds et al. [103], for example, concluded that the “path analysis indicated that the relation between…parenting support and optimal parenting was completely mediated by parenting stress and not by general psychological distress” (p. 409).

The main effects results for the relationships between both the family systems practices and psychological health measures and the three parenting quality measures inform an understanding of which family systems practices and psychological health measures are related to which dimensions of parenting as stated in the call for papers for the Special Issue of the IJERPH on Parenting and Mental Health. This can be ascertained from the results in Table 2, Table 3 and Table 4. First, the effect sizes between the family systems practices measures and the different dimensions of parents’ psychological health are much the same for each of the four family systems practices (Table 2), although the sizes of effect for family needs, family resources, and family strengths are almost twice as large as those for family supports. Second, the family systems practices measures were found to be differentially related to the three parenting quality measures (Table 3), as evidenced by the sizes of effect between measures. For example, the sizes of effects for family needs and family strengths are larger for parenting beliefs compared to parental involvement and parenting practices, whereas the sizes of effect between family resources and supports and the three parenting quality measures are nearly identical. Third, the psychological health measures were found to be differentially related to the three parenting quality measures (Table 4). The sizes of effect for the relationships between general health, depression, stress, well-being, caregiving burden, parenting stress, and parenting beliefs were twice as large as those for parent involvement and parenting practices.

The mediated effects findings also provide the basis for an understanding of the mechanisms for the relationships between intervention-related measures and parenting quality, as described in the call for papers for the special issue of the IJERPH on Parenting and Mental Health. The results from the two sets of mediated analyses (Table 5 and Table 6) indicated that the different types of psychological health measures tended to account for a similar amount of variance between the family systems practices and the parenting quality measures. The total amounts of variance accounted for by the main and mediated effects were also nearly identical for the two sets of mediated analyses.

### Implications for Practice

The results reported in this paper, as well as findings reported elsewhere (e.g., [49,51,54,55,58]), are all consistent with the basic tenets of the applied family systems model [45,46] that guided the conduct of the research. The results provide support for the use of family strengths to obtain both resources and supports to achieve needs satisfaction. Applied intervention studies by the author and his colleagues, where the different family systems model components were operationalized as intervention practices, have all yielded evidence that the model is useful for strengthening family capacity to obtain needed resources and supports. For example, child [105], parent [106], and family [107] strengths have been operationalized as personal interests and abilities and used to both strengthen existing capabilities and promote the acquisition of new competencies . The same has been done with the other family systems intervention practices components. 

## 5. Limitations

Several limitations need to be mentioned to place the results in methodological and procedural perspective. First, the data in all of the meta-analyses are correlational where causal statements may not be warranted. The results reported in this paper simply indicate that there is covariation between the variables of interest where the findings are consistent with hypothesized relationships between the family systems measures, psychological health measures, and the parenting quality measures. Second, there is also the possibility that the obtained relationships between measures were affected by other unobserved variables or by statistical artifacts. These might have resulted in under- or over-estimation of the effect sizes between measures.

There are also limitations related to the use of meta-analysis for aggregating results from different studies. First, combining results from studies that differ for any number of reasons may have resulted in “mixing apples and oranges”. For example, the use of different scales for measuring any one of the parenting quality measures may have resulted in suppression of the strength of the relationships between measures. Second, methodological considerations that were beyond the scope of this paper, such as moderator effects, might explain differences reported in the primary studies. These types of analyses are the next step in the line of research described in this paper.

## 6. Conclusions

The relationships between both the family systems practices and the parents’ psychological health and parenting quality measures were almost entirely direct and independent. The relationships between the family systems practices measures and parenting quality were partially mediated by caregiving burden and parenting stress. Future research should focus on other explanatory variables that might better explain the indirect effects of family systems practices on parenting quality. Previous studies in the line of research described in this paper found that self-efficacy beliefs [48] proved to be a robust predictor of the relationships between family systems practices and parents’ psychological health, family member interactional patterns, parent provision of child learning opportunities, parent-child interactions, and positive parenting practices.

## Figures and Tables

**Figure 1 ijerph-20-06723-f001:**
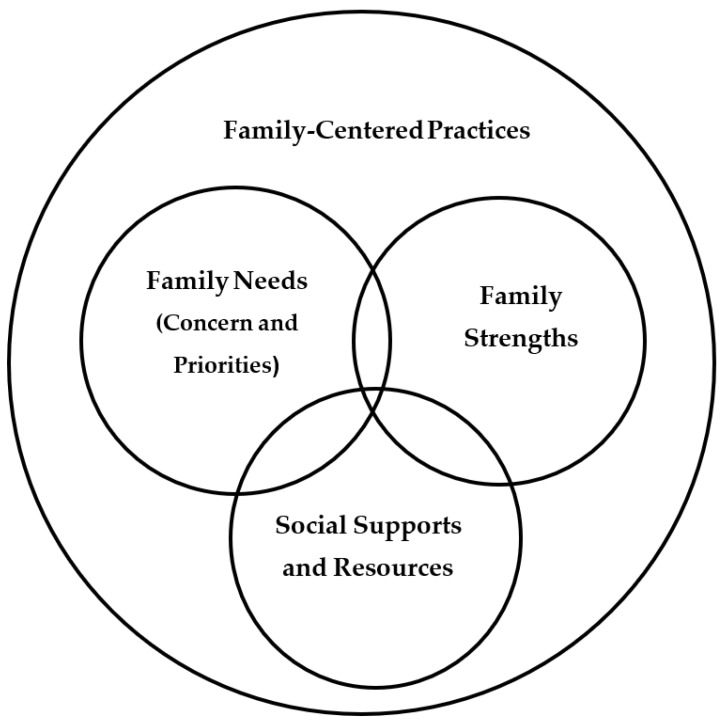
Major components of an applied family systems intervention model.

**Table 2 ijerph-20-06723-t002:** Average weighted effect sizes (r) for the relationships between the family systems practices measures and parents’ psychological health.

Family Systems Measures	k	N	r	95%/CI
Family Needs				
General Health	4	376	0.38	0.25, 0.50
Depression	5	606	0.39	0.33, 0.45
Stress	5	858	0.44	0.17, 0.64
Well-Being	6	573	−0.32	−0.16, −0.47
Parenting Stress	9	1565	0.41	0.30, 0.52
Caregiving Burden	11	2407	0.41	0.30, 0.51
Family Resources				
General Health	13	1429	−0.41	−0.33, −0.48
Depression	14	2837	−0.37	−0.30, −0.44
Stress	13	2699	−0.38	−0.26, −0.50
Well-Being	4	260	0.47	0.15, 0.72
Parenting Stress	20	4170	−0.42	−0.37, −0.47
Caregiving Burden	8	1102	−0.33	−0.24, −0.42
Family Supports				
General Health	28	2301	−0.20	−0.14, −0.26
Depression	30	2967	−0.27	−0.10, −0.42
Stress	13	1022	−0.15	−0.03, −0.27
Well-Being	18	1865	0.33	0.18, 0.48
Parenting Stress	33	5064	−0.22	−0.20, −0.26
Caregiving Burden	15	1253	−0.17	−0.09, −0.24
Family Strengths				
General Health	9	1223	−0.41	−0.33, −0.48
Depression	8	825	−0.43	−0.30, −0.55
Stress	6	1155	−0.23	−0.14, −0.32
Well-Being	10	1693	0.43	0.34, 0.52
Parenting Stress	9	950	−0.42	−0.30, −0.52
Caregiving Burden	6	824	−0.34	−0.13, −0.52

k = Number of effect sizes, N = Number of study participants, r = Average, weighted effect size, and CI = Confidence interval.

**Table 3 ijerph-20-06723-t003:** Average weighted effect sizes (r) for the relationships between the family systems practices measures and parenting quality.

Family Systems Measures	k	N	r	95%/CI
Family Needs				
Parenting Beliefs	6	620	−0.35	−0.28, −0.41
Parent Involvement	6	1465	−0.21	−0.17, −0.25
Parenting Practices	5	1440	−0.19	−0.11, −0.26
Family Resources				
Parenting Beliefs	11	1039	0.24	0.12, 0.35
Parent Involvement	11	1319	0.27	0.18, 0.36
Parenting Practices	14	3294	0.29	0.23, 0.35
Family Supports				
Parenting Beliefs	13	1106	0.22	0.16, 0.28
Parent Involvement	6	1421	0.21	0.15, 0.26
Parenting Practices	7	375	0.17	0.11, 0.24
Family Strengths				
Parenting Beliefs	6	1138	0.44	0.22, 0.62
Parent Involvement	7	661	0.32	0.21, 0.43
Parenting Practices	8	2527	0.36	0.23, 0.48

**Table 5 ijerph-20-06723-t005:** Effects decomposition for the relationships between family systems practices and parenting quality mediated by the parents’ psychological health ^a^.

Family Systems Measures	Direct Effects	Indirect Effects	Total Effects
Family Needs			
Parenting Beliefs	−0.35	−0.15	−0.50
Parent Involvement	−0.21	−0.08	−0.29
Parenting Practices	−0.19	−0.10	−0.29
Family Resources			
Parenting Beliefs	0.24	0.15	0.39
Parent Involvement	0.27	0.08	0.35
Parenting Practices	0.29	0.10	0.39
Family Supports			
Parenting Beliefs	0.22	0.09	0.31
Parent Involvement	0.21	0.05	0.25
Parenting Practices	0.17	0.06	0.23
Family Strengths			
Parenting Beliefs	0.44	0.14	0.58
Parent Involvement	0.32	0.08	0.40
Parenting Practices	0.36	0.10	0.46

^a^ Composite general health, depression, stress, and well-being measures.

**Table 6 ijerph-20-06723-t006:** Effects decomposition for the relationships between family systems practices and parenting quality mediated by parenting stress and caregiving burden.

Family Systems Measures	Direct Effects	Indirect Effects	Total Effects
Family Needs			
Parenting Beliefs	−0.35	−0.16	−0.51
Parent Involvement	−0.21	−0.09	−0.30
Parenting Practices	−0.19	−0.13	−0.32
Family Resources			
Parenting Beliefs	0.24	0.15	0.39
Parent Involvement	0.27	0.08	0.35
Parenting Practices	0.29	0.12	0.41
Family Supports			
Parenting Beliefs	0.22	0.08	0.30
Parent Involvement	0.21	0.04	0.25
Parenting Practices	0.17	0.06	0.23
Family Strengths			
Parenting Beliefs	0.44	0.15	0.59
Parent Involvement	0.32	0.08	0.40
Parenting Practices	0.36	0.12	0.48

## Data Availability

The data for ascertaining the relationships between the psychological health and parenting quality measures are included in the Appendix A for this paper.

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
