# Peer review of "Meta-Analyses of the Relationships between Family Systems Practices, Parents’ Psychological Health, and Parenting Quality"

_ijerph, 2023, doi:10.3390/ijerph20186723_

Round 1

Reviewer 1 Report

Introduction

The author should clarify the difference between parenting behaviors and parenting practices.

Please review the term “family system practices”. The family needs and strength are not family practices.

Please clarify ¿The review is about the effects of interventions focused on family system practices or the relationships of these variables? 

Discussion

Please include the study limitations

Please include a section about theoretical and practical implications of the study findings

Author Response

Thank you for your comments and requests. The following are the edits and additions made in response to your feedback.

  1. The terms parenting behavior and parenting practices are used interchangeably in the literature to describe this aspect of parenting quality. I edited the paper and refer only to parenting practices to reduce any confusion about the terms.
  2.  I added additional text to the Introduction to describe how the family systems model components "together" constitute family systems practices.
  3.  Test was added to the Introduction and Methods and Material sections of the paper to explain that the meta-analyses in my paper include measures of the relationships between the variables of interest.
  4. A section has been added to the paper that describes the Limitations of the types of meta-analyses described in my paper.
  5. A subsection was added to my paper that describes and illustrates the implications of the results for intervention purposes.

Reviewer 2 Report

This is clearly an important area of research and appears to be focussed on bringing together previous research by the author. At times the paper seems to rush through previous findings and could be improved by elaborating e.g. lines 148 - 157 could be improved with more specific information. 

The author states that the operationalisation of determinants is a focus on applied family systems however the discussion focusses a lot on reiterating results and does not provide information regarding the application or interpretation of the results. This could be improved. 

While this is further analysis of the authors work there appears to be an excess of self citations. 

Author Response

Thank you for your comments and suggestions. The following are the ways in which your recommendations were addressed.

  1. The Introduction was edited and additional text was added that further described the family systems model and how the methods described in my paper "ties together" the model components. I also added explanatory text to provide examples of specific information that was requested.
  2. I added a subsection to the paper that provides a description of how the results have informed the operationalization of the family systems intervention model practices to link the research findings to practical application.
  3. I understand that my paper includes many citations from previous publications by the author and his colleagues but we are the only researchers that have engaged in a line of integrated research to test basic tenets of the applied family systems model.

Round 2

Reviewer 2 Report

Thank you for attending to revisions.